# Capsaicin’s Role in Mitigating Muscle Soreness and Enhancing Futsal Players’ Recovery After Exercise-Induced Muscle Damage

**DOI:** 10.3390/nu17050813

**Published:** 2025-02-26

**Authors:** Mina Rashki, Mohammad Hemmatinafar, Kousar Safari, Babak Imanian, Rasoul Rezaei, Maryam Koushkie Jahromi, Katsuhiko Suzuki

**Affiliations:** 1Department of Sport Sciences, Faculty of Education and Psychology, Shiraz University, Shiraz 84334-71946, Iran; 2Faculty of Sport Sciences, Waseda University, 2-579-15 Mikajima, Tokorozawa 359-1192, Japan; katsu.suzu@waseda.jp

**Keywords:** capsaicin, exercise-induced muscle damage (EIMD), delayed-onset muscle soreness (DOMS), futsal players, performance recovery

## Abstract

**Background:** Capsaicin, known for its antioxidant and antibacterial properties, may mitigate oxidative stress and inflammation associated with exercise-induced muscle damage (EIMD). This study evaluates the efficacy of capsaicin supplementation in reducing delayed-onset muscle soreness (DOMS) and enhancing strength and power in collegiate male futsal players. **Methods:** A randomized, double-blind, placebo-controlled crossover design was used. Twelve male futsal players participated in three testing sessions: baseline (BL), followed by capsaicin (12 mg) or placebo (PLA), with a 7-day washout period. Participants consumed the supplement 45 min before completing an EIMD protocol of 200 plyometric jumps with a 10% body-weight vest. Metrics including vertical jump height (VJH), pressure pain threshold (PPT), thigh circumference (TCM), and isokinetic and isometric strengths were assessed 48 h post-EIMD. DOMS was measured using a visual analog scale (VAS) at baseline and 12, 24, and 48 h post-EIMD. **Results:** Capsaicin supplementation significantly improved VJH (*p* = 0.009), PPT (*p* = 0.004), and reduced TCM (*p* = 0.015) compared to baseline, with superior performance in PPT and TCM reduction (*p* < 0.05). Capsaicin also significantly decreased VAS scores for DOMS immediately, 12, 24, and 48 h post-EIMD (*p* < 0.001) compared to PLA and BL. No significant differences were found in isokinetic and isometric strength metrics (*p* > 0.05). **Conclusions:** Acute capsaicin supplementation can mitigate DOMS and enhance performance markers such as VJH and PPT in collegiate futsal players. Its benefits suggest a viable nutritional strategy for recovery and performance optimization in high-intensity sports.

## 1. Introduction

Futsal, a high-intensity indoor sport, demands frequent accelerations, decelerations, rapid directional changes, and repeated jumping, often resulting in significant eccentric muscle contractions [1]. These actions often lead to substantial eccentric muscle contractions, increasing the risk of exercise-induced muscle damage (EIMD). EIMD is characterized by mechanical stress, inflammation, and oxidative stress, which collectively impair muscle function, increase soreness, and reduce athletic performance [2]. A critical manifestation of EIMD is delayed-onset muscle soreness (DOMS), which is associated with elevated muscle protein leakage (e.g., creatine kinase and myoglobin), inflammatory responses, and reactive oxygen species (ROS) production [3]. These factors negatively impact muscle recovery and athletic performance, highlighting the importance of effective interventions to mitigate these effects.

Nutritional strategies have been proposed to address EIMD and DOMS, including anti-inflammatory and antioxidant supplements. Capsaicin, the active component of chili peppers, has garnered attention for its unique physiological effects [4]. Capsaicin activates the transient receptor potential vanilloid 1 (TRPV1) channels, which play a role in pain modulation, anti-inflammatory responses, and mitochondrial biogenesis [5]. TRPV1 activation leads to an increase in intracellular calcium levels, which can enhance metabolic processes and mitochondrial activity, resulting in improved adenosine triphosphate (ATP) production [5]. Additionally, capsaicin’s anti-inflammatory effects are mediated through the inhibition of pro-inflammatory cytokines, such as interleukin-6 (IL-6) and tumor necrosis factor-alpha (TNF-α), and the reduction of oxidative stress markers [6,7]. These mechanisms suggest capsaicin’s multifaceted role in performance enhancement and recovery. Moreover, capsaicin’s ergogenic properties extend to vascular function improvements, such as increased nitric oxide (NO) bioavailability, which enhances blood flow and oxygen delivery to active muscles during exercise [5]. This contributes to reduced muscle fatigue and improved endurance. Capsaicin has also been shown to enhance lipid metabolism and reduce perceived exertion, further supporting its role as a recovery aid [8].

Despite its theoretical benefits, evidence of capsaicin’s effect on DOMS and recovery following EIMD remains limited [8]. Previous studies have demonstrated capsaicin’s ability to reduce pain perception and enhance exercise performance in trained athletes [8]. However, the mechanisms underlying these effects and their practical applications in competitive sports require further exploration. Understanding capsaicin’s role in recovery could offer significant advantages for athletes, particularly during congested training or competition schedules, where optimizing recovery is critical.

While several studies have explored the effects of capsaicin on pain modulation and exercise performance, there is a notable lack of research addressing its comprehensive role in performance enhancement and recovery. Existing literature often focuses on acute pain reduction, with limited investigation into its sustained impact on recovery metrics such as strength restoration, soreness attenuation, and functional performance following high-intensity exercise [9]. Furthermore, most studies are conducted on general or recreational populations, with scarce data available for elite or collegiate athletes engaged in sports like futsal that involve significant eccentric loading. This gap underscores the need for targeted research to evaluate capsaicin’s potential as a dual-action supplement for recovery and performance optimization in such contexts.

Therefore, this study investigates the effects of acute capsaicin supplementation on DOMS, strength, and performance markers in collegiate male futsal players following an EIMD protocol. Using a randomized, double-blind, placebo-controlled, and crossover design, the research aims to evaluate capsaicin’s efficacy in mitigating muscle soreness, improving pain thresholds, and enhancing recovery metrics. The findings of this study will contribute to the growing body of knowledge on sports nutrition and recovery, providing practical insights into capsaicin’s potential applications in high-intensity sports.

## 2. Methods

The participants were recruited from the Shiraz Premier Men’s Futsal League, and the study was conducted in July and August 2024 in the Sport Sciences Department laboratory of Shiraz University. Twelve male futsal players with an average of three years of experience in futsal leagues willingly participated in this study. According to the McKay et al. system, the subjects were placed in Tier 3 (Highly Trained/National Level) [10]. Initially, 14 players were recruited, but two were excluded from continuing due to cold symptoms during the intervention. Table 1 contains the demographic information of the participants. Participants self-reported, through health and exercise history questionnaires, that they possessed a minimum of three years of experience playing futsal, had no history of allergy to capsaicin, and consistently obtained 7–8 h of sleep within 24 h. During the data collection phase, participants indicated that they did not engage in smoking, alcohol consumption, or the intake of caffeine-containing beverages. Before the implementation of the intervention, the study procedures were thoroughly explained, and written informed consent was obtained from each participant. This study followed the Declaration of Helsinki and received approval from The Research Ethics Committees of the Faculty of Psychology and Educational Sciences, Shiraz University (Code: IR.US.PSYEDU.REC.1403.051, 2024). Furthermore, all participants were enrolled in the same training camp and followed identical training regimens under the guidance of their trainers. Participants were also instructed to refrain from strenuous exercises 48 h before and after the intervention sessions. This study was conducted one month before the futsal league season, ensuring relevance to pre-season conditioning and recovery strategies.

### 2.1. Sample Size Calculation and Study Design

The sample size for this study was determined using G*Power analysis software (version 3.1.9.7) [11], with a 5% Type I error rate (α), 0.80 statistical power (1-β), and a correlation of 0.90 between repeated measures, which is typical for crossover designs. The effect size (Δ of response) was estimated to be 0.6, based on a previous capsaicin study [8] that reported this effect size for the total repetition variable. Based on these parameters, the initial calculation suggested a required sample size of 10 participants. However, to account for potential participant dropout and ensure the robustness of the results, the sample size was increased to 12 participants. Variability measures used in the calculation included a standard deviation (SD) of 1227.4 kg for total weight lifted, as reported in the study by Conrado de Freitas et al. (2018) [8]. This measure represents the total mass lifted across all sets, which showed a significant improvement in the capsaicin condition compared with placebo (capsaicin: 3919.4 ± 1227.4 kg vs. placebo: 3179.6 ± 942.4 kg, *p* = 0.002). While total weight lifted is a commonly used metric for evaluating strength performance, isokinetic strength was selected as the primary outcome for this study due to its ability to assess muscle function and recovery across different movement velocities more precisely. This sample size ensures adequate statistical power for detecting significant differences and minimizing the risk of Type II errors.

This study utilized a randomized, double-blind, placebo-controlled, and crossover design to evaluate the effects of capsaicin supplementation on exercise recovery and performance metrics in collegiate male futsal players, following established guidelines for sports nutrition research [4,8] (Figure 1). Twelve participants with at least three years of competitive futsal experience were recruited and randomly assigned to receive either 12 mg of purified capsaicin or visually identical placebo (12 mg starch) in a double-blinded manner. To ensure consistency and minimize inter-individual variability, participants attended two familiarization sessions. They were introduced to all testing procedures, including Biodex dynamometer strength tests and the EIMD protocol, adapted from previous literature [2,12]. Baseline measurements of vertical jump height (VJH), pressure pain threshold (PPT), thigh circumference (TCM), and isokinetic and isometric strength (flexion and extension at 60°/s and 180°/s angular velocities) were recorded prior to supplementation. Participants ingested their assigned supplement 45 min before performing the EIMD protocol, which consisted of 200 maximal vertical jumps with a 10% body-weighted vest [13]. DOMS was assessed using a visual analog scale (VAS) at baseline, immediately post-exercise, and 12, 24, and 48 h after EIMD [14]. At 48 h post-EIMD, measurements of VJH, PPT, TCM, and strength metrics were repeated. After a 7-day washout, participants crossed to the alternate condition and repeated the protocol to eliminate carryover effects. This design ensured high methodological rigor and reliability by controlling for individual differences and standardizing testing conditions, warm-ups, and dietary intake across sessions.

This study’s primary outcomes were DOMS and performance recovery metrics, including isokinetic and isometric strength (e.g., peak torque values for knee extensors and flexors). These measures directly evaluated the effects of acute capsaicin supplementation on muscle soreness and strength recovery following EIMD. The secondary outcomes included PPT and TCM, which assessed pain sensitivity and potential swelling around the thigh. These secondary metrics helped provide a comprehensive assessment of the effect of capsaicin supplementation on muscle function and recovery, complementing the primary outcome measures.

### 2.2. Randomization Process of the Crossover Design

To randomly assign 12 participants to the two different conditions over two weeks, a Latin Square Design [15] was utilized to ensure that each participant experienced each condition exactly once, thereby controlling for order effects. First, all subjects participated in the baseline measurements, and after completing all the tests, they were randomly shuffled and then divided into two groups of six people. Each group was then assigned to the conditions according to a balanced rotation pattern: in Week 1, Group 1 experienced Condition PLA, and Group 2 experienced Condition CAP. In Week 2, the conditions were rotated such that Group 1 moved to Condition CAP and Group 2 to Condition PLA. This approach ensured complete counterbalancing, minimizing potential order effects while maintaining randomization integrity.

### 2.3. Supplementation Procedures

Based on the preceding research [4,16], participants were randomly assigned to consume either a placebo (PLA, 12 mg of starch) or 12 mg of purified capsaicin (CAP; COSMED MANIPULATES^®^, Pedro Leopoldo, Brazil). This dosage was selected because prior studies reported that 12 mg supplementation did not increase gastric motility [8]. Supplements were consumed 45 min before the EIMD protocol. To minimize variability, all trials were conducted between 8 and 11 AM. Participants were closely monitored for any side effects, and no adverse reactions were reported for either CAP or PLA supplementation.

One hour before exercise testing, participants were provided a standardized breakfast containing 350–400 kcal (64% carbohydrates, 20% protein, 16% fats) [17]. This macronutrient breakdown was selected to ensure consistent energy availability for the exercise session. The carbohydrate content was emphasized to provide sustained energy, while protein supported muscle function, and fats helped maintain satiety. This standardized approach aimed to minimize potential confounding effects from varying dietary intake, ensuring consistency across trials and allowing for a more precise assessment of the effects of the tested supplementation. Also, they were instructed to maintain a consistent diet throughout the trial period and avoid eating one hour before testing. Additionally, participants were provided with a list of foods containing capsaicin or chili peppers, such as spicy sauces, chili powders, or hot peppers, and were instructed to avoid these foods 48 h before each session to prevent any potential interference with the intervention. Similar dietary precautions have been reported in previous capsaicin studies to ensure accurate measurement of the supplement’s effects [4,8]. Participants were also asked to abstain from caffeine and strenuous activities 24 h prior to each testing session.

### 2.4. Exercise-Induced Muscle Damage Protocol

Before the EIMD protocol, the participants performed a 10-min warm-up of dynamic movements, slow running, and stretching exercises. Thereafter, participants completed 200 vertical jumps with weighted vests (equivalent to 10% of body weight) to induce muscle damage. For this purpose, each person performed ten sets of 20 maximum jumps (one jump every 4 s). Two minutes of rest (sitting in a chair) were included between sets. To ensure the program’s rigidity, the participants immediately assessed their rate of perceived exertion (RPE). The EIMD protocol was adapted from previous literature on dietary supplementation [12,13]. During the study, all participants were members of the same training camp, and their training protocol was the same under the supervision of trainers.

### 2.5. Examination of Delayed-Onset Muscle Soreness by the VAS Scale

The VAS scale measured the amounts of DOMS. On this scale, a horizontal line of 10 cm is drawn, at the beginning of which the phrase is painless and at the end of which the word is severe pain. VAS is a number that allows a person to express the severity of their pain and is used in experimental and clinical studies. The subjects determined their perception of the severity of DOMS before the start of the test (VAS-before), immediately (VAS-immediately), 12 (VAS-12 h), 24 (VAS-24 h), and 48 (VAS-48 h) hours after EIMD [18]. By measuring the distance of points marked from the line’s origin with a ruler, the pain score of each person was recorded in centimeters [14]. These time points were chosen based on established research indicating that muscle soreness typically begins within 12 h of high-intensity eccentric exercise, peaks around 24–72 h, and gradually resolves after that [19]. This framework allowed for a comprehensive assessment of the progression and peak of DOMS following the intervention.

### 2.6. Pressure Pain Threshold (PPT)

The study determined the PPT using a blood pressure cuff at the midpoint of the femur (Blood Pressure Cuff-Thigh-Double Tube, MDF Instruments, Agoura Hills, CA, USA). The participants were seated on a chair with their knees bent at a 90-degree angle. A 2.5 cm diameter and 25 cm length plastic tube was placed around the femur midline of the dominant leg. The blood pressure gauge cuff was placed around the participant’s thigh and uniformly inflated. The investigator recorded the pressure level at the onset of pain as the PPT in mmHg [12,20].

### 2.7. Thigh Circumference Measurement (TCM)

The perimeter of the femur was measured three times, ensuring no folds were created in the skin, to assess the degree of TCM. A tape measure was utilized, recording measurements to the nearest millimeter. The mean values obtained were documented as the swelling score around the femur. Specific landmarks were marked on the dominant leg to identify the femur’s midpoint accurately. At the same time, the participant was in a standing position, including the greater trochanter of the femur and the tibial prominence [12,21].

### 2.8. Vertical Jump Height (VJH)

The Sargent Jump test was used to measure VJH. The participants chalked the end of their fingertips. Then, participants stood at the shortest distance from a wall and, keeping both feet on the ground, reached up as high as possible with one hand and marked the wall with the tips of their fingers (M_1_). From a static position, the participants jumped as high as possible and marked the wall with the chalk on their fingers (M_2_). VJH was the distance between M_1_ and M_2_. The test was repeated three times with a one-minute passive rest between each attempt, and the largest distance was taken for analysis [22].

### 2.9. Isokinetic and Isometric Strength Tests

An isokinetic dynamometer (System 4 Pro, Biodex Medical Systems, Inc., Shirley, NY, USA) was used to measure the isokinetic strength of the knee extensor and flexor muscles (concentric phase, at an angular velocity of 60°/s and 180°/s, con/con ratio, dominant leg) with five consecutive repetitions in the direction of extension-flexion in both two angular velocity and 60 s of rest for recovery between each set. Gravity correction of the torque measurements was accomplished using the Biodex software package (version 4.X). For the tests, participants were stabilized with straps across the chest, above the knee, around the waist, and above the ankle. This arrangement secured the lower leg to the input shaft of the dynamometer. Furthermore, the estimated transverse rotational axis of the knee was visually aligned with the mechanical axis of the dynamometer. The range of motion of the knee joint during the test was set at 80°, and absolute peak torque (APT), relative peak torque or peak torque per body weight (RPT), and average rate of force development (AvRFD) (AvRFD was calculated using the APT/time to peak torque equation) were measured [23,24]. The selected velocities were identified to facilitate an evaluation of the following parameters: 60°/s: This lower velocity assesses maximal strength under controlled conditions, offering valuable insights into knee extensors’ and flexors’ peak torque production capabilities. Such measurements are critical for ensuring stabilization during soccer-specific activities. 180°/s: This increased velocity evaluates dynamic muscle performance and reflects the functional capacity for executing rapid movements, including sprinting and sudden directional changes encountered during soccer matches [23,25].

Maximum voluntary isometric contraction (MVIC) of the dominant leg was measured at 45° and 60° in away (extension) and toward (flexion) action, using the same device. The isometric testing consisted of 5 maximal efforts for 5 s at the knee angles of 45° and 60° [26]. The Maximum Voluntary Isometric Contraction (MVIC) angles of 45° and 60° were chosen based on their biomechanical relevance and reliability for assessing knee extensor and flexor strength. These angles optimize torque production by aligning the lever arm of the knee joint with muscle fiber orientation, ensuring consistent and reproducible measurements. Additionally, mid-range angles like 45° and 60° closely mimic functional positions during sports-specific movements, such as accelerations and changes in direction, which are common in futsal. Previous studies have demonstrated the validity of these angles for evaluating muscular performance and recovery in dynamic athletic contexts [26].

### 2.10. Data Analysis

Data were analyzed using the statistical package for social sciences (SPSS version 26, Chicago, IL, USA). The data distribution normality was determined using the Shapiro-Wilk test. One-way repeated measures ANOVA test was used to determine the main effect on isokinetic and isometric indicators and functional test results, and the Bonferroni post hoc test was used to determine pairwise differences. The partial eta squared (pEta^2^) was calculated as an effect size measure for interaction and main effects. According to Cohen, pEta^2^ ≥ 0.01 indicates small effects, pEta^2^ ≥ 0.059 medium effects, and pEta^2^ ≥ 0.138 large effects [27]. Also, the percentage of changes compared to the BL (%CV) was calculated through the following formula [28] (%CV = ((PLA or CAP − BL)/|BL|) × 100). The level of statistical significance was *p* ≤ 0.05, and data are presented as mean ± SD. Figure production was also performed using GraphPad Prism (version 9.0.0, GraphPad Software, San Diego, CA, USA).

## 3. Results

Descriptive characteristics are reported in Table 2 and Table 3.
nutrients-17-00813-t002_Table 2Table 2Means and Standard Deviation (Mean ± SD) of the DOMS in each condition (n = 12).TimePLA(cm)CAP(cm)MDSig95% CIMean ± SDMean ± SDVAS-before0.11 ± 0.390.08 ± 0.28−0.030.744−0.19–−0.144VAS-immediately4.97 ± 1.822.86 ± 2.17−2.110.001−2.70–−1.51VAS-12 h3.88 ± 1.802.36 ± 2.03−1.520.001−2.19–−0.858VAS-24 h3.55 ± 1.731.75 ± 1.76−1.800.001−2.47–−1.13VAS-48 h2.36 ± 1.530.58 ± 0.84−1.780.001−2.30–−1.25PLA: Placebo, CAP: capsaicin, MD: Mean Difference, CI: Confidence Interval, VAS: Visual Analog Scale, before: Before the EIMD, immediately: Immediately after the EIMD, 12 h: 12 h after the EIMD, 24 h: 24 h after the EIMD, 48 h: 48 h after the EIMD.
nutrients-17-00813-t003_Table 3Table 3Means, Standard Deviation (Mean ± SD), and %CV of the measured variables in the three conditions.VariablesBLPLA%CV_PLA_CAP%CV_CAP_VJH (cm)
43.00 ± 6.26

44.83 ± 7.14

4.2
46.75 ± 7.028.7
TCM (cm)
53.00 ± 4.91

52.41 ± 3.89

−1.1

50.50 ± 4.01

−4.7
PPT (mmHg)
312.91 ± 49.74

321.66 ± 41.52

2.7

350.00 ± 53.21

11.8
MVICflx45 (Nm)
222.35 ± 35.34

214.80 ± 35.00

−3.3

220.10 ± 43.81

−1.0
MVICext45 (Nm)
132.79 ± 38.82

117.53 ± 13.15

−11.3

134.91 ± 38.72

1.5
MVICflx60 (Nm)
320.60 ± 43.59

308.04 ± 47.21

−3.91

315.04 ± 53.66

−1.7
MVICext60 (Nm)
193.10 ± 60.05

168.85 ± 19.47

−12.5

195.10 ± 60.84

1.0
APTflx60 (Nm)
132.85 ± 38.71

117.50 ± 13.14

−11.5

134.91 ± 38.72

1.5
APText60 (Nm)
222.35 ± 35.34

214.80 ± 35.00

−3.3

220.11 ± 43.81

−1.0
APTflx180 (Nm)
101.43 ± 37.35

87.56 ± 21.55

−13.6

86.52 ± 17.02

−14.6
APText180 (Nm)
163.62 ± 64.24

159.35 ± 46.36

−2.6

149.75 ± 26.60

−8.4
RPTflx60 (%)
186.00 ± 63.96

168.85 ± 19.47

−9.2

195.10 ± 60.84

4.8
RPText60 (%)
304.44 ± 63.46

308.04 ± 47.21

1.1

315.04 ± 53.66

3.4
RPTflx180 (%)
136.84 ± 33.53

118.68 ± 27.19

−13.2

124.16 ± 23.16

−9.2
RPText180 (%)
212.67 ± 35.43

211.60 ± 37.69

−0.5

213.94 ± 31.25

0.5
AvRFDflx60 (N/s)
0.30 ± 0.10

0.33 ± 0.10

0.10

0.34 ± 0.12

0.13
AvRFDext60 (N/s)
0.64 ± 0.30

0.53 ± 0.19

−0.17

0.64 ± 0.21

0
AvRFDflx180 (N/s)
0.37 ± 0.17

0.44 ± 0.26

0.18

0.39 ± 0.24

0.5
AvRFDext180 (N/s)
0.85 ± 0.38

0.71 ± 0.28

−0.16

0.67 ± 0.26

−0.21
BL: Baseline, PLA: Placebo, CAP: capsaicin, %CV: percentage of changes, VJH: Vertical Jump Height, TCM: Thigh circumference measurement, PPT: Pressure pain threshold, MVIC: Maximum Voluntary Isometric Contraction, APT: Absolute Peak Torque, RPT: Relative Peak Torque, AvRFD: Average Rate of Force Development, Nm: Newton meter, N/s: Newton per second, cm: centimeter, mm: millimeter.
Figure 1The protocol of taking supplements and performing tests. VAS: visual analog scale, EIMD: exercise-induced muscle damage.
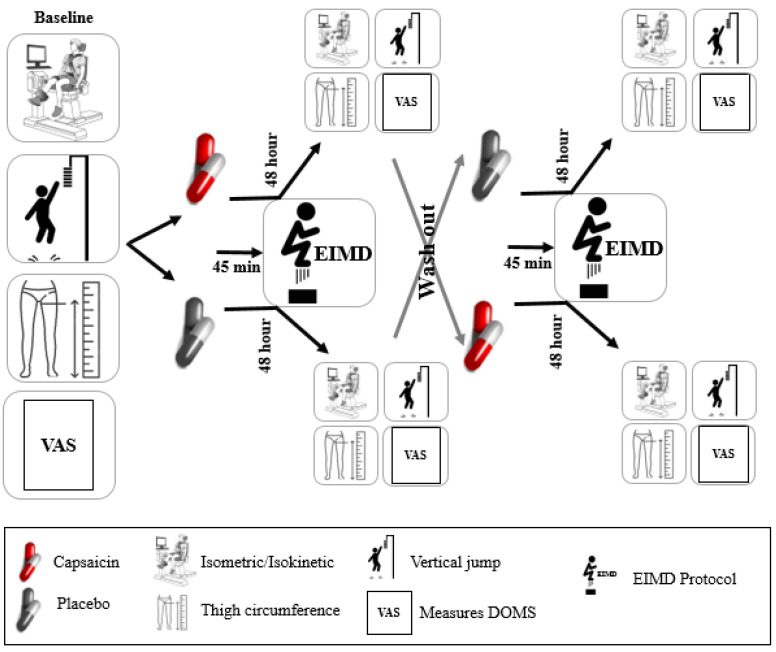


The main effect on DOMS was significant, with CAP significantly reducing VAS scores immediately after EIMD (*p* = 0.001, pEta^2^ = 0.676) compared to PLA and BL (Table 2, Figure 2). Significant reductions in DOMS were also observed in CAP at 12 h (*p* = 0.109, pEta^2^ = 0.217), 24 h (*p* = 0.059, pEta^2^ = 0.287), and 48 h (*p* = 0.001, pEta^2^ = 0.575) post-EIMD. CAP consistently outperformed PLA in reducing DOMS across all time points (Figure 2). 

The results indicated a significant main effect on VJH (F_(1,51)_ = 9.257, *p* = 0.001, pEta^2^ = 0.568). Compared to baseline (BL), both CAP (*p* = 0.009) and PLA (*p* = 0.006) conditions significantly improved VJH scores (Table 3, Figure 3). However, no significant difference was observed between CAP and PLA conditions (*p* = 0.79). The percentage change analysis demonstrated that the CAP condition (8.7%) improved more than the PLA condition (4.2%).

A significant main effect was found for TCM (F_(1,70)_ = 8.982, *p* = 0.001, pEta^2^ = 0.528). Both CAP (*p* = 0.015) and PLA (*p* = 0.007) conditions showed reductions in TCM compared to BL, with CAP demonstrating a greater reduction (−4.7%) than PLA (−1.1%) (Table 3, Figure 3). Capsaicin supplementation produced a significantly greater reduction in thigh circumference compared to placebo (*p* = 0.02).

PPT showed a significant main effect (F_(1,50)_ = 8.812, *p* = 0.002, pEta^2^ = 0.607). Both CAP (*p* = 0.004) and PLA (*p* = 0.008) conditions significantly improved PPT compared to BL, with CAP showing a higher percentage improvement (11.8%) compared to PLA (2.7%) (Table 3, Figure 3). However, no significant difference was observed between CAP and PLA conditions (*p* = 0.42).

**Figure 2 nutrients-17-00813-f002:**
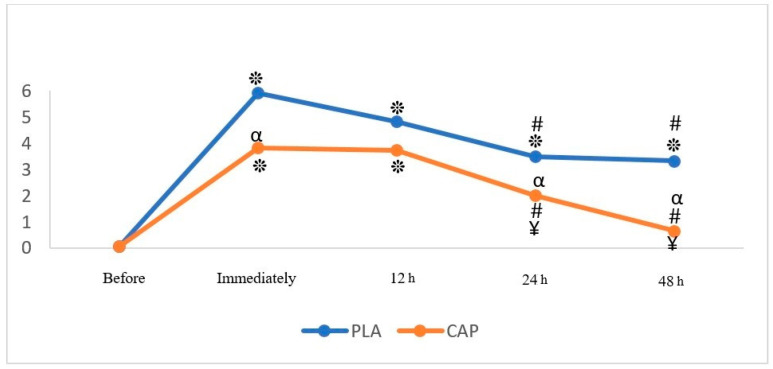
Perceived muscle soreness changes using the VAS scale in PLA or CAP conditions. (PLA: placebo/CAP: capsaicin). *: Significant difference from before, in any condition. #: Significant difference from immediately following, in each condition (*p* < 0.05) ¥: Significant difference from 12 h later, in each condition. α: Significant difference from PLA condition at any time.

**Figure 3 nutrients-17-00813-f003:**
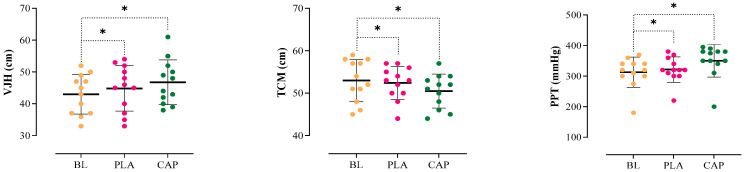
Individual responses, means, and standard deviations of the VJH, TCM, and PPT in the three conditions. BL: Baseline, PLA: Placebo. CAP: Capsaicin, VJH: Vertical Jump Height, TCM: Thigh circumference measurement, PPT: Pressure pain threshold. *: Significant difference compared to the BL (*p* < 0.05).

The results on isokinetic and isometric strength demonstrated that there were no differences in APTflx60 (F_(1,48)_ = 1.391, *p* = 0.270, pEta^2^ = 0.193), APTflx180 (F_(1,53)_ = 1.770, *p* = 0.194 pEta^2^ = 0.132), APText60 (F_(1,61)_ = 0.256, *p* = 0.777, pEta^2^ = 0.060), APText180 (F_(1,71)_ = 0.342, *p* = 0.714, pEta^2^ = 0.004), RPTflx60 (F_(1,25)_ = 0.915, *p* = 0.415, pEta^2^ = 0.090), RPTflx180 (F_(1,81)_ = 3.106, *p* = 0.106, pEta^2^ = 0.311), RPText60 (F_(1,89)_ = 0.135, *p* = 0.874, pEta^2^ = 0.003), RPText180 (F_(1,70)_ = 0.034, *p* = 0.996, pEta^2^ = 0.002), AvRFDflx60 (F_(1,34)_ = 0.813, *p* = 0.417, pEta^2^ = 0.069), AvRFDflx180 (F_(1,80)_ = 0.336, *p* = 0.696, pEta^2^ = 0.030), AvRFDext60 (F_(1,60)_ = 2.573, *p* = 0.113, pEta^2^ = 0.190), AvRFDext180 (F_(1,46)_ = 1.439, *p* = 0.260, pEta^2^ = 0.116), MVICflx45 (F_(1,57)_ = 0.183, *p* = 0.834, pEta^2^ = 0.026), MVICflx60 (F_(1,37)_ = 0.485, *p* = 0.622, pEta^2^ = 0.009), MVICext45 (F_(1,83)_ = 2.530, *p* = 0.140, pEta^2^ = 0.230), MVICext60 (F_(1,91)_ = 2.885, *p* = 0.117, pEta^2^ = 0.079) (Figure 4). All of these results are reported in Table 4.
nutrients-17-00813-t004_Table 4Table 4Pairwise comparisons in the three conditions (n = 12).VariablesPLACAPBLCAPBLPLAVJH (cm)MD1.83−1.913.751.91Sig0.0090.0790.0030.07995%CI0.56–3.09−4.92–0.251.58–5.919−0.25–4.09TCM (cm)MD−0.581.91−2.50−1.91Sig0.3770.0020.0050.00295%CI−1.97–0.810.85–2.98−4.06–−0.932−2.98–−0.851PPT (mmHg)MD8.75−28.3337.0828.33Sig0.2820.0420.0020.04295%CI−8.28–25.78−55.50–−1.1617.29–56.871.16–55.50MVICflx45 (Nm)MD4.702.452.25−2.45Sig0.5990.7850.6860.78595%CI−14.42–23.84−16.88–21.80−9.659–14.15−21.80–16.88MVICext45 (Nm)MD15.42−4.2519.684.25Sig0.0970.6930.400.69395%CI−3.29–34.14−27.35–18.841.030–38.33−18.84–27.35MVICflx60 (Nm)MD−2.99−8.395.408.39Sig0.7630.1190.6100.11995%CI−24.25–18.26−19.32–2.54−17.21–28.01−2.54–19.32MVICext60 (Nm)MD9.57−16.2725.8516.27Sig0.3510.1970.0370.19795%CI−12.05–31.20−42.33–9.781.89–49.80−9.78–42.33APTflx60 (Nm)MD−15.35−17.402.0517.40Sig0.1330.1000.8890.10095%CI−36.19–5.49−38.77–3.95−29.56–33.68−3.95–38.77APText60 (Nm)MD−7.55−5.30−2.245.30Sig0.4220.6960.8230.69695%CI−27.45–12.35−34.41–23.79−23.83–19.35−23.79–34.41APTflx180 (Nm)MD−13.871.03−14.90−1.03Sig0.2220.8730.1240.87395%CI−37.45–9.70−12.89–14.96−34.60–4.78−14.96–12.89APText180 (Nm)MD−4.279.87−14.15−9.87Sig0.8410.5250.4030.52595%CI−50.04–41.49−23.26–43.01−49.99–21.69−43.01–23.26RPTflx60 (%)MD−17.15−26.259.1026.25Sig0.3200.0970.7350.09795%CI−53.38–19.08−58.11–5.61−48.50 –66.70−5.61–58.11RPText60 (%)MD3.60−7.0010.607.00Sig0.8650.7120.6500.71295%CI−42.02–49.22−47.71–33.17−39.48–60.68−33.71–47.71RPTflx180 (%)MD−18.19−5.47−12.715.47Sig0.480.3950.1380.39595%CI−36.18–−0.20−19.09–8.14−30.21–4.77−8.14–19.09RPText180 (%)MD−1.06−2.341.272.34Sig0.8880.8290.8850.82995%CI−17.39–15.26−25.63–20.94−17.60–20.15−20.94–25.63AvRFDflx60 (N/s)MD0.02−0.010.040.01Sig1.0001.0001.0001.00095%CI−0.05–0.11−0.07–0.05−0.07–0.16−0.05–0.07AvRFDext60 (N/s)MD−0.10−0.110.000.11Sig0.4210.1241.0000.12495%CI−0.29–0.08−0.24–0.02−0.13–0.13−0.02–0.24AvRFDflx180 (N/s)MD0.070.040.02−0.04Sig1.0001.0001.0001.00095%CI−0.14–0.28−0.23–0.32−0.19–0.25−0.32–0.23AvRFDext180 (N/s)MD−0.130.03−0.17−0.03Sig0.9491.0000.4341.00095%CI−0.51–0.23−0.17–0.24−0.48–0.13−0.24–0.17BL: Baseline, PLA: Placebo, CAP: capsaicin, MD: Mean Difference, CI: Confidence Interval, VJH: Vertical Jump Height, TCM: Thigh circumference measurement, PPT: Pressure pain threshold, MVIC: Maximum Voluntary Isometric Contraction, APT: Absolute Peak Torque, RPT: Relative Peak Torque, AvRFD: Average Rate of Force Development, Nm: Newton meter, N/s: Newton per second, cm: centimeter, mm: millimeter.
Figure 4Individual responses, means, and standard deviations of the knee flexor and extensor isokinetic and isometric parameters in the three conditions. BL: Baseline, PLA: Placebo, CAP: Capsaicin. MVIC: Maximum Voluntary Isometric Contraction, APT: Absolute Peak Torque, RPT: Relative Peak Torque, AvRFD: Average Rate of Force Development, Nm: Newton meter, N/s: Newton per second.
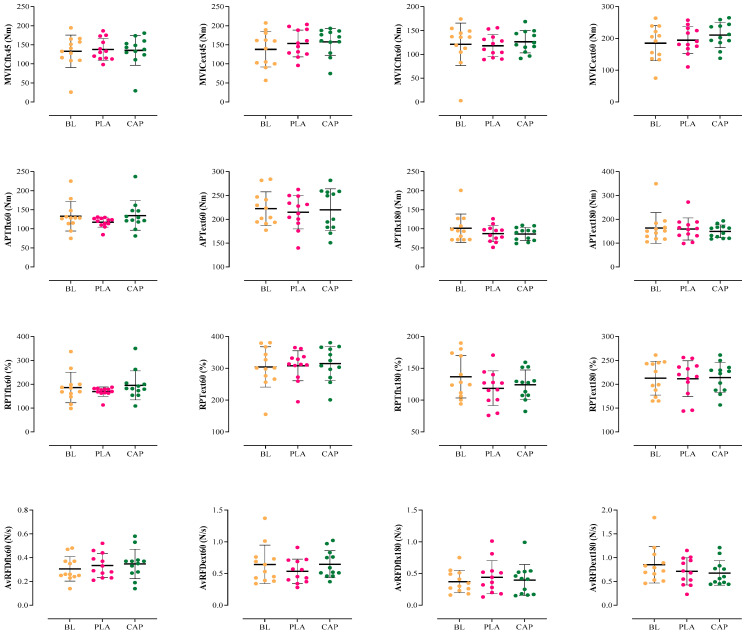


## 4. Discussion

This study investigated the effect of capsaicin supplementation on the markers of muscle damage, inflammation, and reduction of oxidative stress before a traumatic training session on delayed muscle soreness and strength and power performance, using VAS to assess DOMS. PPT test, TCM, VJH, and isometric and isokinetic tests were performed. The results of this study showed that consumption of 12 mg of purified CAP 45 min before the EIMD significantly improves the endurance of the lower body muscles compared to the baseline and PLA and also improves the results of the VAS test, the level of pain perception in PPT decreased, TCM decreased compared to the PLA, the records recorded in Sargent’s vertical test improved, and specific changes were seen in the results of isometric and isokinetic tests. Furthermore, the results showed that CAP significantly reduced DOMS values immediately, 12, 24, and 48 h after EIMD compared to PLA.

### 4.1. Capsaicin’s Role in Reducing DOMS, TCM, and PPT

The observed reductions in DOMS and TCM, along with improvements in PPT, underscore capsaicin’s multifaceted role in modulating pain perception, inflammation, and muscle recovery. Capsaicin exerts these effects primarily through the activation of transient receptor potential vanilloid 1 (TRPV1) channels, which desensitize nociceptors and inhibit the production of pro-inflammatory cytokines such as IL-6 and TNF-α [4,5]. One study investigated capsaicin’s impact on exercise performance, fatigue, and inflammation. The findings suggest that capsaicin supplementation may attenuate the post-exercise rise in interleukin-1β (IL-1β), a pro-inflammatory cytokine, potentially reducing exercise-induced inflammation [4]. Another meta-analysis examined the effects of capsaicin and capsiate supplementation on endurance performance. The analysis revealed that acute supplementation enhances muscular endurance, though its effects on aerobic endurance remain inconclusive. The ergogenic benefits observed may be attributed to capsaicin’s activation of the TRPV1 channel, leading to increased calcium release in muscle cells, elevated fatty acid oxidation, glycogen sparing, and reduced muscular fatigue [29]. These mechanisms alleviate muscle soreness and swelling, as evidenced by the consistent reduction in DOMS across all measured time points (immediately post-EIMD and at 12, 24, and 48 h) and the significant 4.7% decrease in TCM compared to the 1.1% reduction in the placebo group. The substantial decline in TCM highlights capsaicin’s efficacy in addressing muscle swelling, a major contributor to soreness and discomfort. Since muscle inflammation, often mediated by IL-6, serves as both a marker and a regulator of post-exercise recovery [30], capsaicin’s modulation of IL-6 and other inflammatory pathways has been widely demonstrated in both in vitro and in vivo models [5]. Its anti-inflammatory effects are also linked to enhanced mitochondrial function and ATP production, further supporting its role in recovery [8].

The marked reductions in DOMS at all time points align with capsaicin’s analgesic properties mediated through TRPV1 activation, which decreases calcium flux into nociceptive neurons, effectively desensitizing them to painful stimuli [2]. This mechanism likely explains the association between reduced TCM and DOMS, as diminished edema relieves pressure on sensory neurons, resulting in decreased discomfort. Research by Tanabe et al. (2021) corroborates these findings, emphasizing capsaicin’s role in mitigating muscle swelling and enhancing recovery following eccentric exercise [2]. Additionally, the 11.8% improvement in PPT compared to a 2.7% increase in the placebo condition demonstrates capsaicin’s effectiveness in raising pain thresholds by reducing swelling-induced pressure on nociceptors. This interplay between reduced TCM and enhanced PPT highlights the intrinsic connection between inflammation, pain relief, and recovery. By minimizing discomfort and promoting tissue repair, capsaicin not only improves subjective recovery experiences but also enables athletes to resume training more effectively and efficiently.

### 4.2. Functional Improvements in VJH and Strength Metrics

Capsaicin supplementation significantly improved VJH by 8.7%, compared to a 4.2% increase observed in the PLA condition, highlighting its potential to enhance explosive power and neuromuscular performance. This improvement can be attributed to capsaicin’s activation of TRPV1 channels, which regulate Ca^2+^ entry into muscle cells. Elevated intracellular Ca^2+^ levels enhance mitochondrial biogenesis and ATP production, which are critical for sustained muscle contractions and high-intensity efforts [5,8]. Capsaicin also stimulates NO release, inducing vasodilation and improving O_2_ delivery and nutrient supply to active muscle tissues. This enhanced circulation aids in clearing metabolic by-products, accelerating recovery and preparing muscles for subsequent efforts [4]. DOMS reductions and PPT improvements observed with capsaicin likely contributed to the enhanced VJH. The 11.8% increase in PPT, compared to 2.7% in the PLA condition, reflected a higher pain threshold, allowing participants to exert maximal effort. Similarly, the 4.7% reduction in TCM and alleviation of muscle inflammation decreased mechanical and sensory limitations, enabling greater power output. Capsaicin’s analgesic and anti-inflammatory effects likely mitigated inhibitory feedback from afferent III and IV fibers, which are sensitive to nociceptive and metabolic stress during high-intensity tasks [2,31,32,33]. These findings align with research showing that CAP improves time to exhaustion and resistance training performance, particularly after high-intensity intermittent exercise [8]. The enhanced VJH observed in this study is particularly relevant to sports like futsal, where rapid recovery and repeated explosive actions are essential. CAP supports physical recovery and performance by improving DOMS, PPT, and TCM and optimizing energy metabolism.

However, isokinetic and isometric strength metrics showed only modest improvements. For instance, relative peak torque (RPT) improved by 4.1% in the CAP compared to a 1.8% increase in the PLA condition at 60°/s, and MVIC improved by 3.5% in the CAP versus 1.2% in the PLA condition. While these changes did not reach statistical significance, they represent clinically meaningful trends, particularly considering the small sample size (n = 12). The study’s statistical power may have been insufficient to detect more minor but potentially relevant differences in strength metrics, as isokinetic and isometric strength measurements typically exhibit higher variability and require larger sample sizes for robust detection. Moreover, this study’s short duration of supplementation (a single-dose acute intervention) may have limited its ability to induce significant structural recovery, as muscle fiber repair and adaptation typically necessitate prolonged intervention periods and consistent nutrient intake. Chronic supplementation or combining CAP with other recovery strategies, such as protein supplementation or resistance training, may lead to more pronounced improvements in maximal strength [2,6,21,33]. Additionally, individual variability in TRPV1 receptor expression and sensitivity could have contributed to the observed outcomes. Therefore, while the acute CAP supplementation did not significantly improve strength, its notable effects on other recovery metrics, such as DOMS reductions and PPT improvements, suggest its potential role in enhancing overall recovery and performance. These findings underscore the need for future studies with larger cohorts and longer supplementation durations better to assess CAP’s full potential in strength recovery.

### 4.3. Integration with Previous Research

The findings of this study align with and extend upon existing literature regarding CAP supplementation, particularly its anti-inflammatory effects, its potential to enhance recovery and its impact on performance metrics. For instance, Giuriato et al. (2022) highlighted CAP’s role in reducing inflammatory markers such as IL-6 and TNF-α, supporting the reductions in TCM and DOMS observed in this study [4]. CAP’s modulation of these inflammatory pathways has been consistently observed in both in vitro and in vivo models, with a growing body of evidence emphasizing its potential for improving recovery following EIMD [4]. Similarly, de Freitas et al. (2018) demonstrated enhanced performance metrics, including increased muscular endurance and improved recovery times, with chronic CAP supplementation [8]. This suggests that extended use of CAP might lead to additional benefits beyond those observed in this acute supplementation protocol, particularly regarding endurance and metabolic efficiency. Such effects were likely responsible for the improvements in VJH observed in the current study, where CAP supplementation enhanced neuromuscular performance. The lack of significant changes in strength metrics in the present study, such as isokinetic and isometric strength, contrasts with some findings in the literature. For example, Roopashree et al. (2024) found significant effects of sustained capsaicinoid supplementation on endurance and metabolic balance, yet no such improvements were observed in strength recovery in this study [34]. This contrast underscores the complexity of CAP’s effects and the need for further research to clarify its role in strength recovery, particularly given its strong influence on pain reduction and inflammation. e Silva et al. (2024) also demonstrated that, while capsaicinoids contribute to endurance performance, their direct impact on maximal strength recovery remains unclear, suggesting that more extended supplementation periods or complementary strategies may be necessary to leverage CAP’s benefits fully [35].

Further supporting this notion, Sukan–Karaçağıl et al. (2023) conducted a systematic review of randomized controlled studies, noting that while CAP and capsiate can positively affect endurance performance, their impact on strength is more variable [36]. These findings echo the results observed in the present study, where CAP led to notable improvements in performance-related metrics (such as VJH and PPT) but showed limited effects on maximal strength metrics. Chronic supplementation or the combination of CAP with resistance training or protein supplementation might be required to see more significant effects on strength recovery. Furthermore, Grgic et al. (2022), in their meta-analysis on CAP and capsiate, noted that while CAP supplementation enhances endurance performance and fat oxidation, its direct influence on strength performance is more marginal, which aligns with the current study’s findings [29]. This highlights the multifaceted nature of CAP’s effects—while its primary benefits may lie in enhancing endurance and recovery from muscle soreness, its role in maximal strength recovery remains complex and warrants further exploration.

These comparisons demonstrate the varied effects of CAP supplementation, emphasizing its potential to improve muscle recovery and performance through reduced inflammation, enhanced pain tolerance, and improved energy metabolism. However, further studies are needed to investigate the role of chronic supplementation, the combined effects with other recovery strategies, and the potential for personalized approaches based on individual variability in TRPV1 receptor expression and sensitivity.

## 5. Limitations and Future Directions

This study evaluating capsaicin supplementation in mitigating DOMS and enhancing recovery in futsal players has several limitations. First, the small sample size (n = 12) restricts the generalizability of the findings to a broader athletic population. Additionally, the lack of biochemical markers, such as creatine kinase or inflammatory cytokines, limits our ability to comprehensively assess the underlying muscle damage and inflammation mechanisms. The study’s focus on acute supplementation (single dose) may not capture the cumulative effects of chronic capsaicin intake, which could provide more sustained benefits over time. Although improvements were noted in performance metrics such as vertical jump height and pain thresholds, the lack of significant findings in isokinetic and isometric strength metrics suggests that more sensitive measures or longer intervention durations may be needed to observe stronger effects. Moreover, the absence of a female cohort prevents the evaluation of sex-based differences in responses to supplementation, which is an important consideration given potential hormonal influences on muscle recovery. Furthermore, individual variability in TRPV1 receptor expression and sensitivity could have influenced the outcomes, as genetic and physiological differences among participants may have resulted in varied responses to capsaicin supplementation. Future studies should address these factors, considering both genetic and physiological differences, to better understand the full spectrum of capsaicin’s effects on muscle recovery and performance.

## 6. Conclusions

In conclusion, this study provides robust evidence for the efficacy of capsaicin supplementation in reducing DOMS, enhancing PPT and VJH, and decreasing TCM following EIMD. These findings underscore and highlight capsaicin’s potential as a valuable nutritional strategy for improving recovery and performance in athletes. While improvements in isokinetic and isometric strength metrics were modest, the observed trends suggest potential benefits that warrant further investigation. Athletes and practitioners should consider incorporating capsaicin supplementation into recovery protocols, particularly for activities involving high eccentric loading, to facilitate quicker and more effective recovery.

## Figures and Tables

**Table 1 nutrients-17-00813-t001:** Characteristics of participants.

Characteristic	Mean ± SD (n = 12)
Age (year)	21 ± 2
Height (cm)	178 ± 7
Weight (kg)	69 ± 5

## Data Availability

The raw data supporting the conclusions of this article will be made available by the authors on request.

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
