# Peer review of "Capsaicin’s Role in Mitigating Muscle Soreness and Enhancing Futsal Players’ Recovery After Exercise-Induced Muscle Damage"

_nutrients, 2025, doi:10.3390/nu17050813_

Round 1
Reviewer 1 Report
Comments and Suggestions for Authors
Title: Capsaicin’s Role in Mitigating Muscle Soreness and Enhancing Futsal Player's Recovery After Exercise-Induced Muscle Damage
This study presents an interesting investigation into the effects of acute capsaicin supplementation on muscle soreness and recovery in collegiate futsal players. Its strengths include a well-designed randomised, double-blind, placebo-controlled, crossover protocol; a clear presentation of multiple performance and recovery markers (VJH, PPT, TCM, DOMS via VAS, and isokinetic/isometric strength); and thorough statistical analyses. The discussion thoughtfully links the observed outcomes with underlying mechanisms (e.g., TRPV1 activation, anti-inflammatory effects, enhanced mitochondrial function) and places the findings within the context of existing literature.
However, there are several areas that require improvement. Firstly, the sample size (n = 12) is relatively small, a fact which the authors acknowledge, but this limitation should be more strongly emphasised along with suggestions for future research. In addition, there are issues related to clarity, consistency in formatting (particularly with abbreviations, p-values, and figures), and occasional grammatical errors. A more concise writing style in parts of the discussion could help the reader to follow the rationale behind the experimental design and the interpretation of results.
The following issues should be noted:
- The randomised, double‐blind, placebo‐controlled, crossover design is comprehensively described. However, the sample size calculation should include additional details (e.g. assumptions, variability measures) to support the power analysis.
- The manuscript would benefit from a clearer statement of both primary and secondary outcomes in the Methods section.
- While the statistical analyses are comprehensive, the presentation of p-values, effect sizes, and confidence intervals is not always clear (e.g., "F1.51" might be better formatted as "F(1,51)"). The figures and tables are informative but could be improved by standardising the legends, ensuring that units are consistently noted, and removing any line breaks that disrupt words (e.g., "capsacin").
- The discussion appropriately links the findings to potential mechanisms, but some paragraphs are overly long and could be split for clarity. The limitations section should more strongly emphasise the issues of sample size, the acute nature of the supplementation (versus chronic effects), and the need to account for individual variability (e.g., in TRPV1 receptor expression).
Corrections by lines:
Line 2. “Futsal Player's Recovery” should be corrected to “Futsal Players’ Recovery” (if referring to multiple athletes).
Lines 12–14. Consider rephrasing “alleviating delayed-onset muscle soreness (DOMS)” to “reducing delayed-onset muscle soreness (DOMS)” for smoother flow.
Lines 42–43. The description of futsal demands should be tightened for clarity. For example, “Futsal, a high-intensity indoor sport that demands frequent accelerations, decelerations, and rapid directional changes, often results in significant eccentric muscle contractions…”
Lines 55–57. The explanation of capsaicin’s activation of TRPV1 channels is clear, but ensure that all technical terms are defined on first use.
Lines 93–98. In the description of participants, ensure that “7 to 8 hours” of sleep is consistently formatted (e.g., “7–8 hours”).
Lines 119–125. When describing the crossover design, be sure to specify the randomization process clearly.
Line 141. When referencing the supplementation procedure, check that the product name “Cosmed Manipulates®” is consistently spelled and capitalized.
Line 150. “Standardized breakfast” details are given; consider clarifying the macronutrient breakdown with a brief rationale for its selection.
Lines 161–167. The EIMD protocol description is detailed; however, the phrase “presented and assessed the rate of perceived exertion (RPE) scale” could be simplified to “immediately assessed their rate of perceived exertion (RPE).”
Lines 267–273. The reporting of VJH results would benefit from clearer formatting; for instance, use “p = 0.009” rather than “(p = 0.009)” in the middle of sentences.
Tables 2 and 3. Ensure that all abbreviations (e.g., “TCM,” “PPT,” “MVIC”) are defined in the table captions.
Line 295. “CAP also significantly outperformed PLA (P = 0.02) in reducing thigh circumference.” It might be clearer to state, “Capsaicin supplementation produced a significantly greater reduction in thigh circumference compared to placebo (P = 0.02).”
Lines 346–353. There are instances of redundant phrasing. For example, “...following EIMD and DOMS tests by VAS questionnaire” could be shortened to “...using VAS to assess DOMS.”
Lines 360–367. When discussing the anti-inflammatory properties of capsaicin, consider citing additional relevant studies if available.
Lines 410–415. The discussion of the modest improvements in strength metrics would benefit from a comment on potential strategies (e.g., chronic supplementation) for future research. •
General: Look for consistency in terminology (e.g., “EIMD protocol” vs. “exercise-induced muscle damage protocol”).
In summary, the present manuscript addresses a significant topic in the field of sports nutrition and recovery research. Following a series of revisions aimed at enhancing clarity, consistency, and the detailed reporting of methods and results, the manuscript has the potential to make a substantial contribution to the existing literature. It is recommended that the authors address both the global issues noted above and the specific line-by-line corrections to ensure the enhancement of the manuscript's overall readability and impact.
Comments on the Quality of English Language
Please, see report
Reviewer 2 Report
Comments and Suggestions for Authors
I’ve got the opportunity to read and review the paper by Mina Rashki et al. This is a study about the role of capsaicin on regeneration among soccer players. I found this paper interesting and I noted some areas that could be improved. Please find detailed review below.
Major comments:
- Where would you place participants according to the McKay et al. system? See doi: 10.1123/ijspp.2021-0451.
- Double-check whether you addressed all issues raised in the CONSORT guidelines.
- Did you register your study in the clinicaltrials.gov database? If yes, provide the number. If not, you could retrospectively register yet.
- When you measured the blood pressure, did you ensure a resting period before measuring the blood pressure to allow it to stabilize?
- Specify the recruitment period and locations.
Minor comments:
- Remember that key paper about recovery and DOMS should be discussed. See doi: 10.3390/jcm11082077.
- I do not. see the aim to explain the exact statistical methods in lines 25-27. Those lines are not necessary, especially because the methods are long.
- Describe your results more precisely in the abstract and reconsider shortening the methods to fulfill the word limit.
To sum up my report, I recommend revisions. The authors should focus especially on major comments.
Reviewer 3 Report
Comments and Suggestions for Authors
Please see attached

Round 2
Reviewer 2 Report
Comments and Suggestions for Authors
The authors revised their article properly.
